# Framing South Asian politics: An analysis of Indian and Pakistani English print media discourses regarding Kartarpur corridor

**Fasih Ahmed** [ORCID]*, **Muhammad Mubeen, Muhammad Nawaz**

Department of Humanities, COMSATS University, Islamabad, Pakistan

* ahmadfasih33@gmail.com, fasih.ahmed@comsats.edu.pk

## Abstract

This paper aims to analyze the divergent perspectives of Indian and Pakistani English print media on opening the Kartarpur corridor. It is a four-kilometer-long cordoned-off strip from the Indo-Pak international border to the Gurdwara Darbar Sahib Kartarpur located in Pakistan. The basic purpose of establishing this corridor is to give easy access to the Indian Sikh community. The initiative was taken into account in August 2018, which resulted in the appearance of a vast quantity of contemplations in the national print media of both countries, especially until the opening of the corridor in November 2019. Print media plays a key role in building knowledge and framing the general public's opinion through interpreting an issue. The data were taken from *Dawn*, *The News International* (Pakistan), *The Times of India*, and *Hindustan Times* (India) from August 2018 to March 2020 using Lexus Nexus Library. The corpus analysis was carried out by applying the lexical study of Natural Language Processing (NLP) through its Latent Dirichlet Allocation (LDA) tool to find out the general patterns or topics in the print media of both countries. It was found that Pakistani print media terms the Kartarpur corridor as a sign of regional peace, religious tourism, mediation, and diplomatic efforts. In contrast, Indian print media focuses on apprehensions related to traveling modalities, pilgrimage facilities, and tensions between the two states.

## Introduction

The opening of the Kartarpur corridor on the India-Pakistan border made several noteworthy headlines in the daily newspapers of both states. It was planned to be opened in 2019 to facilitate the Sikh community for participating in the 550<sup>th</sup> birth anniversary of Baba Guru Nanak, located about four kilometers away from the border in Narowal district, Pakistan. It was a long-lasting desire of the Sikh community of the Indian Punjab since the Partition of 1947. The proposal required the bilateral consent of the Pakistani and Indian governments. This initiative sparked a long debate on all sorts of media on both sides to rationalize and predict the upcoming developments connected with the issue along with multiple questions, such as, what is Gurdwara Darbar Sahib Kartarpur (Kartarpur Darbar)?; why is it important for the Sikh community?; how did the Partition of 1947 generate the issue of Kartarpur corridor?; and what is the core purpose of the recent initiative carried out in 2018?

**Data Availability Statement:** The raw data can be accessed using this link. https://figshare.com/s/fef11e172da7956b7c27.

**Funding:** The authors received no specific funding for this work.

**Competing interests:** The authors have declared that no competing interests exist.

Kartarpur Darbar is where Baba Guru Nanak Dev Ji, the first Guru and the founder of the Sikh religion, spent the last eighteen years (1522–1539) of his life, where he charted the basic principles of his philosophy. During the ensuing centuries, Kartarpur assumed mammoth worth–especially for Sikhism–as all the three prominent religions of the region converge here [1]. The *gurdwara* inhabits both a tomb and a cremation site (*smadh*) as the description of Guru Nanak's disappearance from the world connects both the rituals performed then [2]. Regarding the Gurdwara Darbar Sahib, the Pakistani Prime Minister Imran Khan described the feelings of Sikhs for their holy places, during inaugurating ceremony of Kartarpur Corridor on November 9, 2019, in the following words: "Kartarpur is your Madina, and Nankana Sahib is your Makkah (the two most sacred spaces in Saudi Arabia for the Muslims). We (Muslims) can't even imagine someone keeping us away from Makkah or Madina." Thus, the Kartarpur Darbar is one of the leading sacred spaces of the Sikh religion after Nankana Sahib (Gurdwara Janam Asthan—built on the site where Guru Nanak was born) is currently located in Pakistan.

The historical perspective of the *gurdwara* is also significant to discuss for this study. During the partition of the sub-continent, the Radcliff Award, the boundary commission to define the international border between the newly formed states of India and Pakistan–left Kartarpur town on the Pakistani side in the wake of the Indo-Pak Partition of 1947. It deprived Sikhs of the eastern Punjab of their unrestricted access to this holy site. However, there are many divergent opinions on the Award's inclusion of Kartarpur town in Pakistan; the most solid one is the Ravi River, located three kilometers away from the Kartarpur Darbar on the eastern side fixed as the dividing point in the region between the states. In addition to the visa restrictions, the partition stretched out the distance from a couple of kilometers to manifold as the only two key border crossings between the two countries, Wagah and Munabao, are quite far from this place in the south.

Moreover, since the days of the partition, the stiffness of relations between the two countries made the access to the Kartarpur Darbar far tougher for the Indian nationals during the post-partition settings [3]. Therefore, the pilgrimage to the Kartarpur Darbar for the Indian Sikhs was being transformed into a scarcely achievable desire for the last couple of generations. It can be quite clearly gathered from the sentiments of Navjot Singh Sidhu in his speech at the time of opening of the corridor that four generations of his forefathers died across the border yearning for this Darbar's visit. Thus, it has been a time-honored wish of the Indian Sikhs to open the access strip up to make it an easy and economic pilgrimage.

The two countries discussed the opening of the corridor several times, in 1998, 2004, and 2008 without valuable results. However, the proposal of opening the Kartarpur corridor for Sikh pilgrims in August 2018, and later official groundbreaking of the project in November 2019 proved to be a new beginning in relations between the two countries, which have fought three major wars and a limited conflict since 1947. The announcement ignited media of both sides and during the following year (August 2018 –March 2020), generating massive data on both sides' media perspectives, which is the main focus of this study. The reconstruction of the *darbar* started soon after the official announcement, and it was completed in a short time of ten months leading from the border straight to the Kartarpur Darbar. The courtyard of the *darbar* was extended from 4 acres to 42 acres. The facilities were upgraded in the complex to accommodate more pilgrims. The expansion includes a new courtyard, museum, library, dormitories, locker rooms, an immigration center, and an embankment to protect the shrine from floods. The corridor also features a new border entry point and a bridge to cross the Ravi River. The initiative is an essential trice in Indian Sikhs' access to their sacred spaces in Pakistan [4].

Nevertheless, before digging into the newspaper data and the application of NLP to extract significant themes promoted in print media of both the countries, the existing researches

relevant to this study are referred here to find out the gaps to be filled in by the current study. This study consulted three types of published researches:

1. Research works published on Sikh sacred spaces in Pakistan and the efforts related to facilitating the Sikh pilgrimage towards Pakistan to comprehend the actual issue of the Kartarpur corridor in between India and Pakistan.

2. Researches published on applying Natural Language Processing (NLP) tools like Topic Modeling on newspaper data.

3. Researches published to explain the theoretical triangulation in the context of media entrapping, agenda-setting, and opinion formation.

Sikh sacred spaces in Pakistan are discussed in many research works; a few are discussed here. Some authors, in the background of the opening of the Kartarpur corridor, have highlighted the importance of holy places of Sikhism in Pakistan, their control and conservation since 1947, and their role in the religious sentiments of the Sikh community; in that way endorsing the need of the opening of the corridor for the Indian Sikhs [4, 5]. Likewise, a British historian of South Asian history refers to discernments of the evolving Sikh nationalism in light of evolving policies from both governments [6]. He further states that the Sikh community has played a role multiple times as a bridge between the two states to initiate dialogue on various collaborations like that of Kartarpur facilitation. Various research studies conclude that opening the Kartarpur corridor, which has been welcomed enthusiastically by the Indian Sikh community, can be a game-changer in bilateral relations between the two countries for sustainable cooperation and durable peace [7–9]. Some other studies appreciated Kartarpur as a step towards religious diplomacy and secular identity in South Asia [10–12].

The second type of research we have consulted to handle this study methodologically is the data published on NLP application and its tools like Topic Modeling and the LDA on the print media corpus. Topic Modelling relates to computational linguistics, where social scientists bring textual datasets that are big in size and scope under consideration to make sense of the large chunk of text by extracting topics, especially in social sciences[13, 14]. The development of LDA as a reliable tool of the textual application of NLP to generate topic modeling for texts Blei, Ng [15]. Hoffman, Bach [16]. This method extracts themes from the data by document modeling, text classification, and collaborative filtering. Some other studies give a more comprehensive view about applying LDA on newspapers' textual data as the newspaper data is a mixture of multinomial various topic decomposition techniques of automatic text processing [17–19]. The use of topic modelling and LDA identify the potential issues of interest about the contents of the data and the matching topics solely by their high probability word lists.

The third angle of the study, the theoretical analysis, is based on theoretical triangulation, the combination of at least two or more theoretical perspectives or approaches or data analysis methods. The conceptual triangulation is traveled from a few studies, in which multiple theories or methodological pluralism–mixed, quantitative, and qualitative–have been employed simultaneously to explain the same phenomenon to reduce, refute, or equalize the deficiency of a single stratagem, thereby expanding the ability to interpret the findings [20–22]. Secondly, various media manipulation-related theories and their applications on content analysis of newspapers texts have been drawn from several authentic types of research to enhance the analysis of the current study in light of the practical examples. Agenda setting theory is appropriate to understand communication dynamics, especially in mass communication [23]. Agenda-setting highlights evolving role of media personnel in shaping public opinion via media projections, on the one hand, by floating the knowledge about any issue and, on the other hand, determining the level of importance of that issue from the amount of information

in a news story; thereby converting media into a marketplace of ideas for the readers [24]. Other works on agenda-setting are also consulted to explore further the dynamics behind such a manipulation [25, 26]. The other model, 'Framing Theory' explicated by McCombs Shaw, is also concerned with studying media content and its effects on the audience McCombs, Shaw [24]. Likewise, in their writings on the sensationalist vs. serious type press's roles in framing the contemporary political and economic dynamics in the western world, some authors have raised the importance of the framing theory [27, 28].

The review of the previous studies provides a clear picture of the studies conducted on the Kartarpur corridor, the efforts and proceedings of the opening of the Kartarpur corridor, and its significance in the region from the perspective of peace and stability. However, from the above review, it is noted that no concrete corpus-based research has thus far been carried out using NLP techniques on the newspaper's data and media framing through the respective national print media regarding the Kartarpur corridor. Therefore, it becomes quite important to investigate how the issue of Kartarpur has been echoed on both sides of the border, which took more than 70 years to open as the media played a key role in framing the concerns and opportunities in the masses on both sides. Generally, the framing of the masses, especially through media, paves the way for governments to plan their course of action, the aspect observed in the mainstream newspapers of both countries concerning their vested interests. The present study in the context of Kartarpur focused on the reflection of the Kartarpur discourses to understand how it was presented in front of the masses keeping in view the interest of their countries. Furthermore, the present study also compares the English print media of India and Pakistan to determine how far the neighboring countries determine their future relations.

## Methodology

The corpus of four English newspapers of India and Pakistan was developed from *Hindustan Times* and *Times of India* (India) and *The News* and *Dawn* (Pakistan). The online database LexisNexis was used to access the newspaper's data (August 2018 to February 2020). The rationale for selecting this period is that the issue was officially raised during the oath-taking ceremony of PM Pakistan. Further developments of construction and discourses emerged in media after that oath-taking ceremony in Pakistan. The data relating to each country was kept separately. The dataset was divided into two main parts; the first part of the data was based on headlines and detailed text. The present study only includes detailed text for the analysis to know the perspective on the Kartarpur corridor. The detailed text provides a clearer picture of the problem and helps better in topic modeling. The analysis of the data is based on a two-step process. Firstly, data processing and application of LDA and, secondly, the data analysis through theoretical triangulation. The corpus based on Pakistani newspapers consists of 770 newspaper articles and the corpus based on Indian newspapers consists 977 articles.

### Data filtration

The process of data filtration involved multiple steps. The first phase involved preprocessing. Preprocessing helps get rid of paralinguistic features [29]. These paralinguistic features generally include hyperlinks, file headers, markups, metadata. They make the data noisy, which later becomes a hurdle to extract the hidden meaning from the textual data. All these features were removed using regular expressions. In the second phase, the process of tokenization was applied. Tokenization is the process of defining the word boundaries and categorizing these boundaries into parts of speech in language [30]. After the tokenization, the process of normalization was followed. The normalization of the text brings all the text to lowercase, and all the

punctuations were removed. Subsequently, the stop words were removed from the data. In natural language processing, the stop words are the words that make less contribution in meaning-making. Generally, stop words include, auxiliaries, etc. The defined numbers of these words are 179.

In the last phase of normalization, the text was lemmatized. The lemmatization converts various forms of the words into the root form or the required forms by the researchers [31]. In the case of the present study, only four basic forms of the text were preserved: nouns, adjectives, verbs, and adverbs.

## Application of LDA

The LDA model was developed by Blei et al. (2003), and it helps in an unsupervised machine learning algorithm that learns the underlying topics of a set of documents. LDA provides several keywords and arranges them based on their weightage in the group. The word having high weightage refers to the influence of that word in the group.

In the first phase, LDA was applied to explore topics for the present study. In the second phase, the keywords in the topics were investigated to get a common theme out of them summarized. Later on, the extracted topics were analyzed in detail to get a clear understanding of the text. For the sake of validity, perplexity and coherence score of LDA topics were also determined. The perplexity score determines how difficult or easier a model is to understand for the algorithm applied on a corpus. In this case, the lower the perplexity score, the better the model comprehension is assumed. In the case of the present study, the perplexity and coherence score of LDA regarding Indian newspapers was -11.79 and 0.39, respectively, while Pakistani newspapers were -10.43 and 0.32, respectively.

The present study applying LDA extracted 20 groups of topics regarding the data of each country. Each group of topics further consisted of 10 keywords. These keywords were analyzed in detail to find out the relevant label or theme. These themes guided in getting a clear picture of the text and the stance adopted by each country's newspapers. Using the LDA statistics provided in Tables 1 and 2, the visualizations were made to make the themes more comprehensible.

## Theoretical framework

Grounded theory was applied for the analysis based on the meaning being grounded in the text without confirming prior hypotheses or assumptions [32, 33]. In other words, it is a data-driven/text mining approach where data are the epicenter and meaning from the data is focused and explained. Theoretically speaking, the media of both sides is divergent in its nature of the production of the data in newspapers, thereby leading to the question 'why divergent perspectives?' leads to a conceptual discussion of media's role in making general understanding of its readers. Thus, theoretical triangulation is employed to dig out the matter deeply. The theoretical triangulation uses multiple theoretical schemes to interpret the same results [22]. The present study triangulated grounded theory, framing and agenda-setting theory.

According to Glaser and Strauss [34], the qualitative research of certain empirical phenomena develops some concepts that are grounded in the data of the topic, and it can also be extracted through inductive methods of research by combining both qualitative and quantitative approaches to apprehend the actual context of the issue under study. On the other hand, framing in media generates meanings by focusing on specific events to locate them within a certain broad understanding. The theory tries to analyze the connection between the framing of certain information and the perception generated by such a framing. So, the cause and effects relationship of the presentation generated by media is the actual object of analysis [24]. Lastly, the agenda-setting theory further digs into the process of perception generation in the

**Table 1. Extracted keywords and proposed themes from Indian newspapers' corpus.**

| No | | Labels |
|----|---|--------|
| 0 | '0.064*"ceremony"+ 0.063*"terror" + 0.061*"lay" + 0.046*"cite" + ''0.042*"attack" + 0.032*"pm" + 0.028*"decline" + 0.026*"come" + ''0.026*"batch" + 0.025*"line''' | Apprehensions |
| 1 | , '0.071*"shrine" + 0.057*"free" + 0.051*"connect" + 0.046*"site" + ''0.046*"announce" + 0.041*"arrangement" + 0.038*"member" + 0.036*"team" + ''0.034*"delegation" + 0.033*"share'''), | Emphasis on free entrée |
| 2 | '0.047*"issue" + 0.042*"make" + 0.039*"take" + 0.034*"country"+ "0.027*"could" + 0.027*"add" + 0.026*"peace" + 0.024*"ask" + "0.023*"decision" + 0.017* "islamabad''', | States Relation |
| 3 | '0.223*"sign" + 0.176*"passport" + 0.073*"hand" + 0.054*"away" + "0.041*"waiver" + 0.040*"document" + 0.023*"levy" + 0.019*"sgpc" + "0.009*"online registration"+ 0.005*"many''' | Tour Facilities |
| 4 | '0.154*"travel" + 0.048*"cross" + 0.048*"official" + 0.043*"last" + ''0.035*"offer" + 0.034*"jatha" + 0.032*"due" + 0.029*"claim" + ''0.027*"arrive" + 0.025*"month''' | Travelling Plan |
| 5 | '0.100*"last" + 0.072*"week" + 0.067*"year" + 0.065*"inaugurate" + ''0.060*"tell" + 0.049*"neighbour" + 0.035*"create" + 0.031*"accept" + ''0.025*"history" + 0.025*"reply''' | Invitation for Inaugural Ceremony |
| 6 | '0.142*"tension"+ .096*"several" + 0.051*"mark" + 0.048*"mean" + ''0.048*"bajwa" + 0.033*"nation" + 0.028*"post" + 0.026*"do" + ''0.022*"building"+0.021*"celebrate''' | The Rivalry Between Two Countries |
| 7 | '0.168*"leader" + 0.139*"activity" + 0.066*"gurdwara" + 0.061*"force" + ''0.034*"undertake" + 0.020*"focus" + 0.016*"aspire" + 0.012*"prisoner" + ''0.007*"show" + 0.004*"bus''' | Gurdwara as a Goodwill Gesture |
| 8 | '0.103*"minister" + 0.096*"respect" + 0.071*"help" + 0.057*"remark" + "0.037*"war" + 0.037*"comment" + 0.034*"googly" + 0.032*"advise" +"0.029*"initiative"+0.028*"sentiment''' | Respect Sentiment for the Initiative |
| 9 | '0.095*"meeting" + 0.071*"hold" + 0.068*"official" + 0.054*"expect" + ''0.043*"side" + 0.043*"discuss" +0.037*"modality" + 0.030*"delegation" + "0.028*"issue" + 0.028*"talk''' | Modalities |
| 10 | '0.143*"work" + 0.086*"side" + 0.067*"construction"+ 0.050*"complete" + ''0.042*"detail" + 0.037*"project" + 0.024*"certain" + 0.023*"review" + ''0.022*"end" + 0.021*"hour''' | Project Report |
| 11 | '0.115*"link" + 0.066*"believe" + 0.064*"ensure" + 0.055*"date" + ''0.054*"condition"+ 0.048*"celebration" + 0.042*"set" + 0.036*"already" + ''0.030*"propose" + 0.028*"important''' | Safety and Security of Pilgrims |
| 12 | '0.172*"agreement" + 0.084*"yet" + 0.046*"chawla" + 0.043*"include" + ''0.041*"final" + 0.032*"body" + 0.028*"fail" + 0.027*"appoint" + "0.027*"presence"+ 0.026*"agency''' | Indian Concerns about Disputed Personalities |
| 13 | '0.100*"pilgrim" + 0.066*"nanak" + 0.060*"indian" + 0.046*"open" + ''0.046*"visit" + 0.045*"government" +0.044*"guru" + 0.038*"allow" + ''0.035*"sikh" + 0.027*"day''', | Pilgrims' Access to Darbar |
| 14 | '0.179*"gurdaspur" + 0.142*"land" + 0.075*"district" + 0.045*"acquire" + "0.016*"notification" + 0.015*"object" +0.008*"perform" + 0.008*"intention" ''+ 0.006*"intend" + 0.005*"warm''' | Acquisition of Land |
| 15 | '0.105*"people" + 0.048*"time" + 0.037*"founder" + 0.030*"especially" + "0.026*"clear" + 0.023*"remain" + 0.019*"try" + 0.018*"open" + 0.017*"face" ''+ 0.017*"official''' | People Commitment |
| 16 | '0.080*"statement" + 0.061*"start" + 0.059*"meet" + 0.058*"action" + ''0.046*"announcement"+ 0.041*"highway" + 0.036*"medium" + 0.029*"public" + ''0.027*"advance" + 0.023*"interaction''' | Emphasis on Completion |
| 17 | '0.101*"historic" + 0.075*"move" + 0.063*"charge" + 0.053*"early" + ''0.052*"impose" + 0.043*"write" + 0.039*"service" + 0.037*"passage" + "0.036*"put" + 0.020*"partition''' | Critique on Service Charges by Pakistan |
| 18 | '0.109*"singh" + 0.097*"sidhu" + 0.075*"go" + 0.062*"attend" + ''0.047*"navjot" + 0.043*"tie" + 0.043*"visit" + 0.040*"invite" + "0.035*"event" + 0.035*"former''' | Sikh leader in Developing Relationship |
| 19 | '0.072*"permission" + 0.069*"require" + 0.065*"ceremony" + ''0.059*"inauguration" + 0.048*"recent" + 0.046*"tweet" + 0.046*"political" + ''0.039*"could" + 0.034*"list" + 0.030*"permit''' | Inauguration Permission |

audience by focusing on the behavior of the audience (which is also framed in the information floated by the media) to choose and accentuate the information from media to generate their own opinion [35, 36]. Thus, all the theories mentioned above are employed in the present study to understand the role of media across the border in opening the Kartarpur corridor. The rationale for making theoretical triangulation is that one theoretical stance is not fulfilling all results. Hence, selecting several theories better fit the study's scope [22].

## Interpretation of results with analysis

The data analysis is divided into two parts, Indian print media and Pakistani print media. First the results and analysis of Indian Newspapers are provided in the following:

**Table 2. Extracted keywords and proposed themes from Pakistani newspapers.**

| No | Keywords | Themes |
|---|---|---|
| 0 | '0.064*"relation" + 0.042*"region" + 0.041*"peace" + 0.039*"bilateral" + "0.034*"improve" + 0.033*"situation" + 0.031*"believe" + 0.025*"dialogue" + "0.021*"time" + 0.021*"terrorist´´´ | Regional Peace |
| 1 | '0.100*"express" + 0.081*"nation" + 0.052*"future" + 0.046*"claim" + "0.033*"corridor" + 0.032*"cooperation" + 0.028*"follower" + 0.028*"follow" ''+ 0.024*"fulfil" + 0.022*"tweet´´´ | Cooperation |
| 2 | '0.037*"include" + 0.027*"already" + 0.027*"diplomatic" + 0.026*"case" + "0.026*"dialogue" + 0.024*"continue" + 0.023*"decision" + 0.020*"announce" + "0.019*"member" + 0.018*"tie´´´ | Diplomatic Efforts |
| 3 | '0.067*"sikh" + 0.056*"government" + 0.055*"visit" + 0.046*"pilgrim" + "0.039*"community" + 0.037*"religious" + 0.029*"make" + 0.028*"open" + "0.026*"world" + 0.019*"promote´´´ | Religious Tourism |
| 4 | '0.050*"peace" + 0.035*"country" + 0.032*"government" + 0.031*"minority" + '0.027*"open" + 0.021*"want" + 0.019*"state"+0.018*"bring"+ 0.017*"take" '+ 0.017*"come´´´ | Religious Harmony |
| 5 | '0.045*"project" + 0.039*"war" + 0.035*"let" + 0.028*"live" + ' 0.027*"possible" + 0.026*"perform" + 0.022*"circumstance" + ' 0.021*"completion" + 0.021*"leadership" + 0.019*"understand´´´ | Leadership Perspective from War to Peace |
| 6 | '0.053*"indian" + 0.020*"government" + 0.017*"invite" + 0.017*"issue" + '0.016*"make" + 0.015*"take" + 0.014*"pakistani" + 0.014*"offer" + "0.013*"side" + 0.012*"never´´´ | Stepping for Friendship |
| 7 | '0.029*"effort" + 0.028*"process" + 0.027*"issue" + 0.026*"talk" + ' 0.022*"indian" + 0.015*"help" + 0.015*"peace" + 0.014*"support" + "0.013*"engage" + 0.013*"continue´´´ | Indian Positive Response |
| 8 | '0.058*"concern" + 0.050*"area" + 0.043*"meeting" + 0.040*"psgpc" + "0.040*"medium" + 0.034*"report" + 0.032*"seek" + 0.030*"appoint" + '0.022*"convey" + 0.019*"flag´´´ | Indian Concerns |
| 9 | '0.000*"petition" + 0.000*"mediator" + 0.000*"unite" + 0.000*"revoke" + "0.000*"realty" + 0.000*"enhance" + 0.000*"communitie" + 0.000*"indiaheld" + '0.000*"truly" + 0.000*"mediation´´´ | Mediation |
| 10 | '0.053*"appreciate" + 0.044*"point" + 0.031*"meet" + 0.030*"propose" + "0.029*"technical" + 0.029*"mean" + 0.026*"source" + 0.019*"meanwhile" + '0.018*"representative" + 0.017*"full´´´ | Goodwill Gesture by International Community |
| 11 | '0.044*"rule" + 0.041*"announce" + 0.034*"inauguration" + 0.032*"allow" + '0.025*"day" + 0.025*"sentiment" + 0.025*"pilgrim" + 0.022*"announcement" + '0.020*"founder" + 0.020*"give´´´ | Pilgrims' Sentiments |
| 12 | '0.084*"border" + 0.078*"spokesman" + 0.070*"pakistan" + 0.067*"policy" + "0.050*"terrorism" + 0.042*"hold" + 0.024*"reiterate" + 0.023*"friendship" + '0.020*"regard" + 0.020*"reject´´´ | Terrorism as a Threat to Friendship |
| 13 | '0.056*"open" + 0.032*"go" + 0.030*"election" + 0.028*"indian" + "0.021*"measure" + 0.020*"talk" + 0.018*"decision" + 0.018*"attend" + "0.018*"demand" + 0.017*"welcome´´´ | Transitional government decision |
| 14 | '0.035*"nanak" + 0.029*"attend" + 0.029*"sidhu" + 0.028*"singh" + "0.026*"government" + 0.025*"stand" + 0.022*"yatree" + 0.022*"soon" + "0.021*"ceremony"+ 0.021*"inauguration´´´ | Inaugural Ceremony |
| 15 | '0.038*"year" + 0.026*"respect" + 0.026*"anniversary" + 0.025*"side" + "0.025*"last" + 0.023*"history" + 0.021*"religion" + 0.020*"expose" + "0.019*"month" + 0.016*"inaugurate´´´ | Baba Guru Nanak 550th Anniversary |
| 16 | '0.117*"tourism" + 0.053*"stand" + 0.045*"however" + 0.036*"great" + ' 0.034*"plan" + 0.031*"imran" + 0.028*"status" + 0.025*"must" + ' 0.023*"complete" + 0.021*"find´´´ | Religious Tourism as a Mainstream Policy |
| 17 | '0.052*"open" + 0.044*"help" + 0.042*"door" + 0.037*"set" + 0.037*"sikhs" + ' 0.031*"sikh" + 0.029*"visitor" + 0.027*"nanak" + 0.026*"easily" + "0.023*"thank´´´ | Compliments of Sikh Community |
| 18 | '0.038*"side" + 0.027*"true" + 0.024*"serious" + 0.024*"business" + ' 0.023*"indian" + 0.019*"meet" + 0.017*"technical" + 0.016*"punjabis" + '0.016*"support" + 0.015*"session´´´ | Business Opportunity |
| 19 | '0.043*"positive" + 0.030*"step" + 0.024*"take" + 0.023*"create" + "0.022*"add" + 0.021*"term" + 0.021*"development" + 0.021*"region" + '0.017*"forward" + 0.017*"place´´´ | Positive Step Towards Regional Development |

## Interpretation of results with analysis based on Indian newspapers corpus

Table 1 shows the topics extracted from the data regarding Indian newspaper corpus relevant to the Kartarpur corridor.

Table 1 exhibits the extracted keywords of Indian print media in the left column. Based on the keywords, the themes are generated and the key features of the topics through their relevancy with one another to get the overall trend of the news. The table explains 20 groups of keywords along with the suggested topics next to them. These keywords have been arranged in the groups based on their weightage. The keyword having higher weightage shows a stronger influence in the group. The topics on the left-hand side of the table present a summarized picture of the relevant group of the keywords.

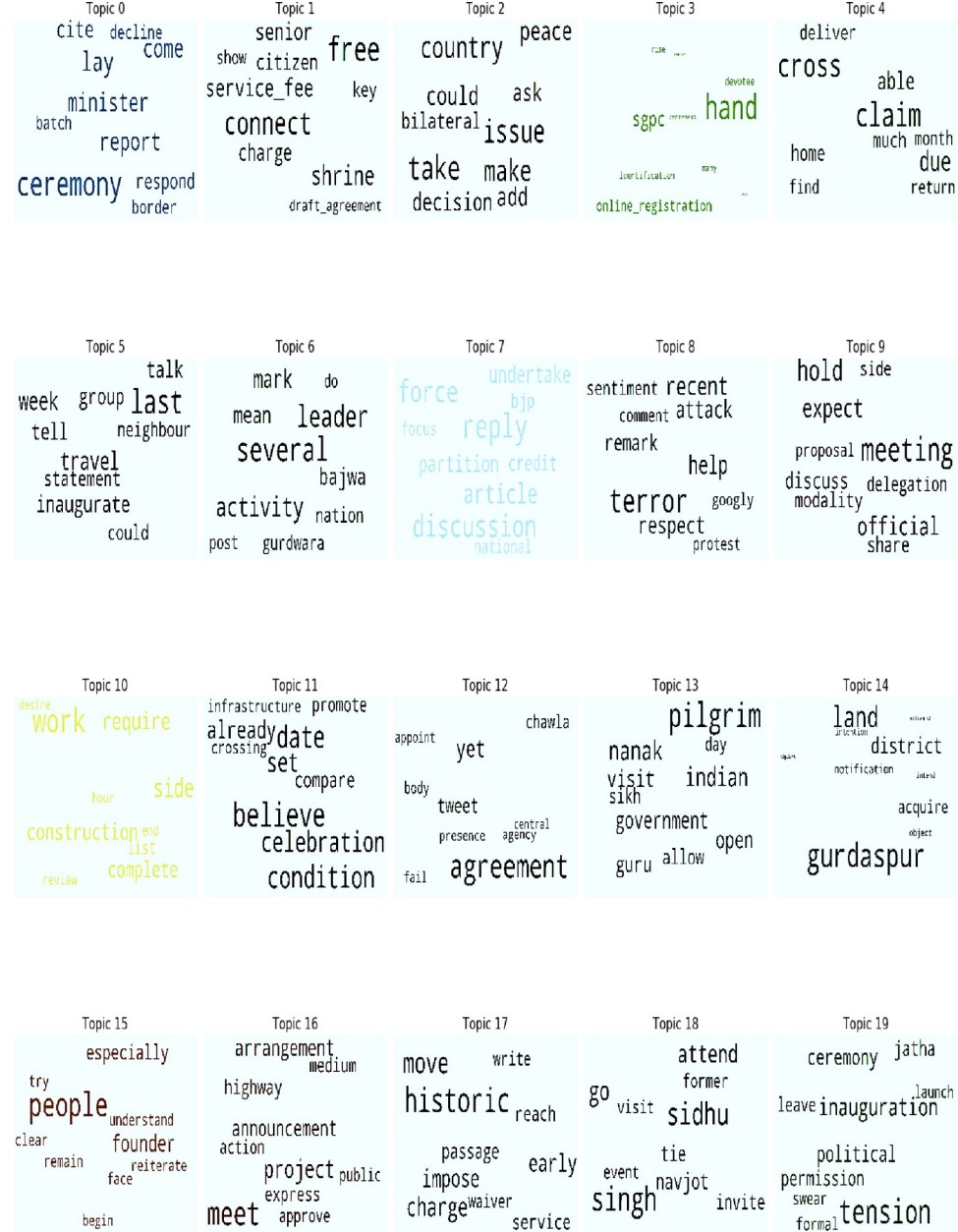

**Fig 1. Word clouds extracted from Indian newspapers.**

The topics in Table 1 (i.e., topic 13, topic 2, topic 9, topic 0, topic 6, topic 11, topic 12, topic 17) relate to pilgrimages access to *darbar*, state relations, modalities, apprehensions, the rivalry between the two nations, safety and security of pilgrimages, and critique on services by charges fall under the category of concerns displayed in Indian newspapers. On the other hand, topics in Table 2 (i.e., topic 4, topic 5, topic 1, topic 3, topic 10, topic 14, topic 16, topic 19) relate to preparations to materialize the project on the Indian Government.

Topics explained in Table 1 have also been shown through word a cloud in Fig 1 which makes the topics and themes clearer. The word cloud of each topic is based on the weightage

number associated with each keyword, as explained in Table 1. As a result, the difference in the size of the keywords shows the high weightage number in Fig 1.

The word clouds help to visualize and understand the topics. It is evident that some of the keywords in Fig 1 are not shown in Table 1. The keywords bigger in size in the word cloud guide about the topic of the keywords such as topic 0 and topic 1 in Fig 1 validate the labels related to apprehensions and emphasis on free entry by the Indian government. Similarly, topic 2 relates to bilateral relations between the two countries as labelled in Table 1, and topic 3 is about tour facilities and online registration systems for pilgrims. Topic 4 relates to the traveling plan or how many pilgrims can visit on monthly bases. Topic 3 and 4 mostly relate to procedural matters. Topic 5 relates to the discussion about the inaugural ceremony on both sides of the border. The rivalry between the two countries has been highlighted in topic 6, resulting in strenuous relations between both the neighbors. Topic 7 presents *Gurdwara* as a goodwill gesture between the two countries. Topic 8 relates to the level of respect for the initiative taken by both the countries as a joint adventure to normalize the situation. Topic 9 relates to the modalities set for corridor. Similarly, topic 10 relates to the project report regarding the construction of the corridor on both sides and topic 11 relates to the safety and security of the pilgrims. The concerns raised by Indian side regarding the disputed personalities have been summed up in topic 12 as this may hinder the overall peace initiative. Topic 13 and 14 relate to the pilgrims' access to *darbar* and acquisition of land for the project under consideration. Topic 15 and 16 relate to people's commitment and emphasis on completion. Critique on service charges has been shown as topic 18. The Indian concern was that it should be charged free whereas the Pakistani side maintained charges for pilgrims. The last topic relates to the matter of the inauguration ceremony and the participation of some political personalities in it.

The extracted topics have been visualized to show the relevancy, variation, and distance between the extracted topics in Fig 2.

Fig 2 displays an inter-topic distance map via multidimensional scaling of the first 20 topics of the data. The relevance metric presents the top-30 most relevant terms for topic1, ranging from the most to the least frequent, and shows 31.4% of tokens. The blue color refers to the

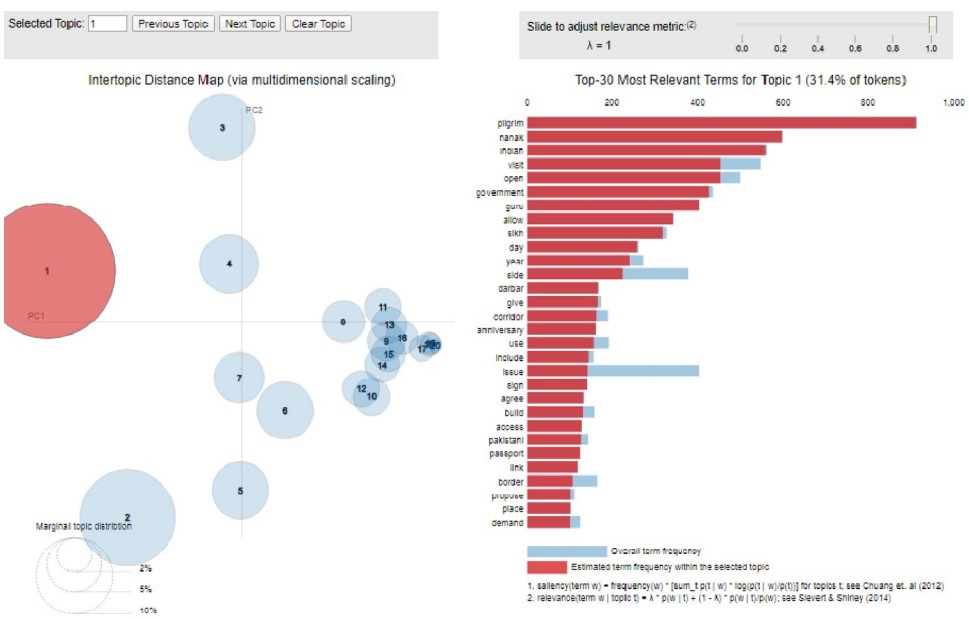

**Fig 2. Topics extracted from Indian newspapers.**

overall term frequency, and the red color shows the estimated term frequency within the selected topics. The relevancy variation and distance between the topics show how different or similar the topics are. On this basis, the extracted topics can be divided into two categories. The topics which are completely distant show that they consist of less variation and are relevant to single topic. On the other hand, Fig 2 also shows that some of the topics overlap with other topics. This implies that one topic can have multiple themes, and some themes are found common between these overlapping topics. From Fig 2, it is evident that topic 1, 2, 3, 4, 5, 6, 7, and 8 do not overlap with any other topic. It means that these topics only deal with a single theme. On the other hand, topic 9, 10, 11, 12, 13, 14, 15, 16, 17, 18, 19, and 20 overlap with one other. This implies that these topics share some themes with other overlapping topics.

## Interpretation of results with analysis based on Pakistani newspapers corpus

LDA was also applied to Pakistani newspapers. The data were extracted from the two leading newspapers. Table 2 explains the range of topics reflected in the Pakistani newspaper corpus. Table 2 consists of 20 groups, and each group of keywords further consists of 10 keywords. These keywords are arranged in LDA based on the higher weightage shown next to each keyword. This group of keywords also consists of suggesting topics in front of them.

Table 2 consists of a group of keywords along with the suggested topics. The range of topics can be categorized into some general topics. The topics (i.e., topic 3, topic 4, topic 16, topic, 18, topic 19) relate opening of the Kartarpur corridor as part of religious tourism. From this perspective, the Pakistani print media frames it as an opportunity for prosperity. The topics (i.e., topic 0, topic 2, topic 5, topic 6, topic 12) fall under the umbrella of peace and development. Topics 14, 15, and 16 in Table 2 relate to the preparations being made on the Pakistani side. Lastly, topic 8 relates to the concerns raised by the Indian Government.

Topics explained in Table 2 are shown in the form of word cloud in Fig 3 to make the topics clearer.

The word clouds help in reaching at a clear understanding of the topic. In Fig 3 topic 0 relates to regional peace where bilateral dialogue can overcome the threat of terrorism. Topic 1 relates to the aspect of cooperation between the two neighbours. The size of the words in word clouds suggests that their cooperation may improve their bilateral ties in future. Topic 2 relates to the focus on continuous diplomatic efforts to improve the ties. Topic 3 relates to the religious tourism as it can be beneficial for both countries. Hence, it is an opportunity for both countries to boost religious tourism as there are multiple religious' places of Muslims, Hindus and Sikhs. The aspect of religious harmony is reflected in topic 4. The size of keywords in Fig 4 suggests that minorities in both countries to be given freedom for their religious practices. It would promote religious harmony and respect for each other on both sides of the border. Topics 5 and 6 take a shift from war to peace between the two countries as it is the only possible solution to move forward. Topic 7 relates to the Indian positive response on the matter of the Kartarpur Corridor where the Indian government is resolved to move forward on the project between the two countries. However, the Indian concerns regarding this project have also been discussed in topic 8 which implies that there is a need to build the trust between the two neighbors. Topics 9 and 10 relate to mediation and the welcoming gestures by the world community on the project of Kartarpur Corridor. Topic 11 relates to pilgrims' sentiments to visit their religious place. Topic 12 focuses on the threat which may jeopardize this project or peace and harmony between the two countries. Topics 13 and 14 center on the transitional government's decision regarding the inaugural ceremony of the project. Topics 15 and 16 relate to 550[th] anniversary of Baba Guru Nanak and religious tourism as an opportunity which may

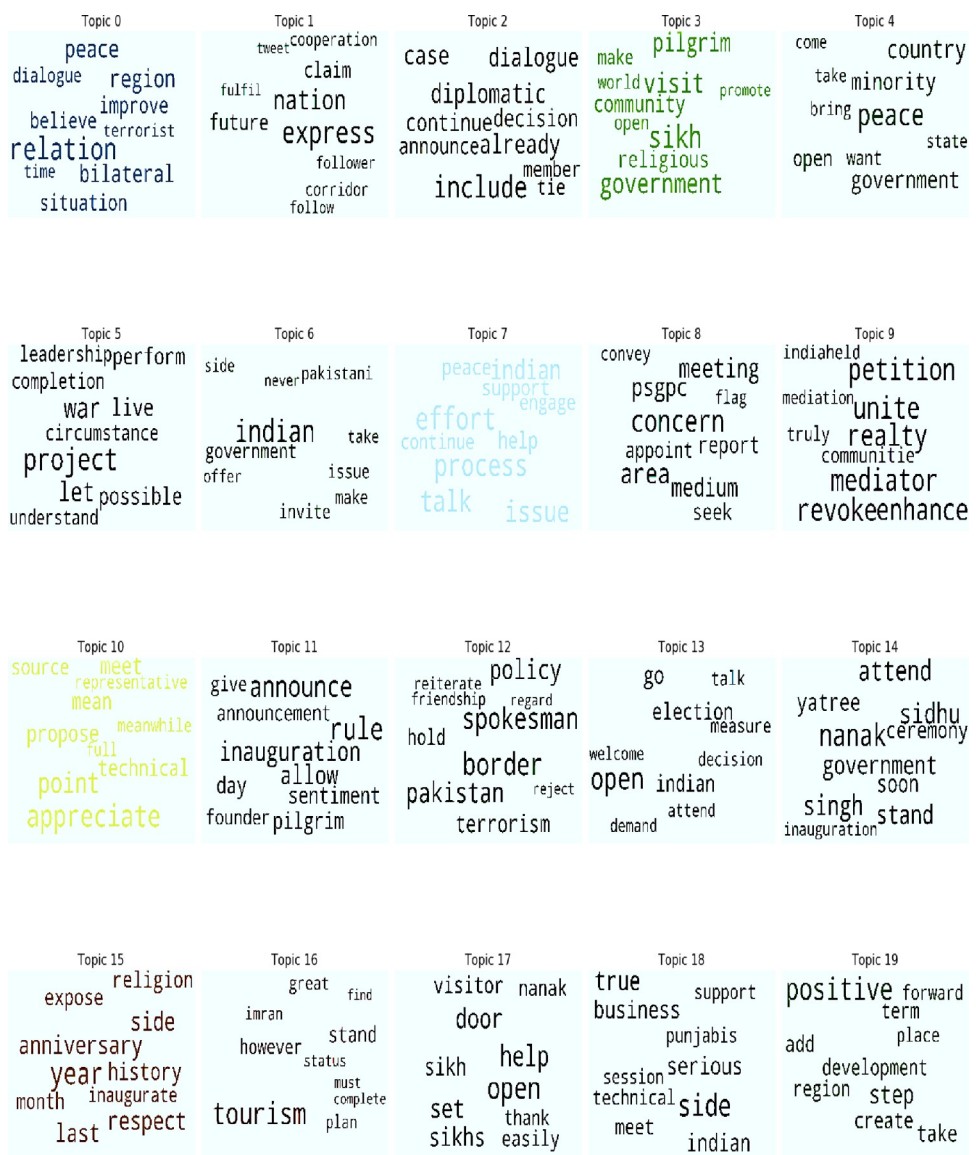

**Fig 3. Word clouds of topics extracted from Pakistani newspapers.**

break the ice in the relationship between the two countries. Topic 17 highlights the welcoming sign of Sikh community as the completion of this project may end their long awaited journey towards the Kartarpur Corridor. The Kartarpur corridor has also been seen as a business opportunity in topic 18 and a positive step in regional development from the perspective of the both countries in topic 19 as well.

Fig 4 exhibits an inter-topic distance map via multidimensional scaling of the most frequent topics of Pakistani English media (1–20 topics). The second part of the figure displays the top-30 most relevant terms of the topic ranging from the most to the least for topic 1, with 15.7% tokens showing overall term frequency in blue color and estimated term frequency in red color within the selected topics. Topics extracted through LDA have also been visualized. Fig 4 shows the arrangement and distance between the topics. Some of the topics in Fig 2 have been shown as overlapping with one another, such as topic 2, topic 3, topic 4, and topic 5. It reflects

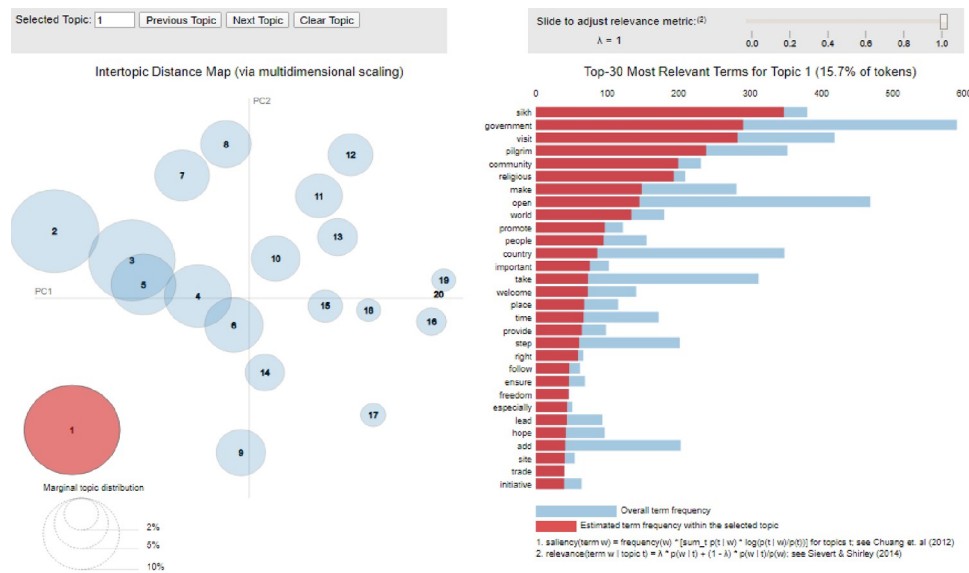

**Fig 4. Topics extracted from Pakistani newspapers.**

that these topics share themes with one another. On the other hand, there are topics which do not overlap showing that these topics contain single theme as compared to the overlapping topics.

## Discussion

The study investigated the viewpoints of Pakistani and Indian newspapers based on the Kartarpur corridor opening. The newspaper data regarding the Kartarpur corridor was retrieved from August 2018 to March 2020. Overall, the outlook of the generated themes by Pakistani newspapers may be summarized by combining the words like "regional peace", "cooperation", "religious harmony," "550th birthday of Guru Nanak", "bilateral relations", "bilateral trade", "religious tourism", "Sikh pilgrims' sentiments," "business opportunities", "mediation and diplomatic efforts", and "positive Indian response" etc.

Contrary to the Pakistani media perspective, the analysis exhibits the various groups of words generated through the LDA of the Indian print media (August 2018-March 2020). The focus of the Indian print media is generally noticed on topics like "traveling plans", "traveling modalities", "Indian government's policy towards Sikh pilgrims", "pilgrimage facilities", "visa concerns", "tensions between the two states", "Indian concerns on PSGPC", "project report", "project land" and "project completion" etc.

The topics extracted and framed above, modeled by the LDA application, are grounded in the data generated by the print media of both the countries whose quantitative depiction is clear from Tables 1 and 2. Qualitatively speaking, the generated topics give us a clear-cut contextual production of the media hubs on the issue. It shows that both sides highlight their understanding of the Kartarpur corridor and their respective concerns. Based on the findings, it can be ascertained that the Indian print media highlights procedural concerns, thereby framing its audience to get more and more concerned about the technical issues of the project while dealing with a rival neighboring state. The topics highlight that their concerns are based on their previous experience with the neighboring state due to procedural matters gaining more importance in their media [37]. Thus, it can be deducted from the topics that Indian

media focuses more on concerns and modalities to complete the project while giving preference to regional peace.

In contrast, keeping in view the same grounded theory, the themes grounded in the media production of the Pakistani side depict an image-building course through the Kartarpur corridor. Moreover, themes projected through Pakistani media reflect the government's plan to drive the nation toward religious harmony, tolerance, and religious tourism [38]. Thus, it can be inferred that Pakistani media portrays it as an opportunity to enhance religious tourism and religious harmony.

The generated themes by topic modeling highlight that the media of both sides has generated certain frames to back their respective stance on the issue. The themes (based on the frequency of words) emphasize their deportment to be floated and projected to influence the perceptions of the intended readership. By emphasizing only the apprehensions and procedural matters, the Indian media generates a way of looking into its audience. So, the general understanding affected by the media would leave certain problematic concerns regarding the Kartarpur corridor with Pakistan. On the other side, as evident from the themes, Pakistani media is affecting its audience through its frames with an ideal role of the Pakistani government for the Sikh community and an approach towards regional peace.

Applying the attribute agenda-setting theory even takes us further to the limitations generated by print media of both sides for their audience that work as blinders in a way. For instance, the Indian print media, in their depictions, on the one hand, restricts the overall intentions and outcomes of the Kartarpur corridor project, especially for the Sikh community, and, on the other hand, highlights the problem-oriented concerns, practicalities, and technicalities. The very agenda limits its audience to think of the project in a mixed way. On the Pakistani side; Pakistani media, on the one hand, restricts on generating any historical background and reasons for the decades-long delay to the project despite the presence of protocol on visits to a religious shrine between the two states and promotes only the face-value glorification of the Kartarpur project thereby confining its audience to remain inline of the state narrative to develop any perception regarding the Kartarpur project.

## Conclusion

Conclusively, it can be determined from the above discussion that print media is a key player in floating knowledge on the opening of the Kartarpur corridor. The debate that started after the announcement of the initiative of opening the corridor generated perceptions based on the perspectives generated and highlighted by the print media of India and Pakistan that are quite divergent. Based on results and discussion, the Pakistani print media sees the Kartarpur corridor as an opportunity for religious tourism and religious harmony. There is very little focus on the concerns.

On the other hand, the Indian print media raises concerns as primary agenda compared to other project modalities. Hence it can be concluded that the agenda of both media is quite divergent. The natural language processing with its lexical tools, topic modeling, and LDA is quite helpful in determining the nature of the text produced in print media on the Kartarpur corridor issue, which is not quite common in the context of South Asian academic research. By hypothesizing the debate on perspectives, it becomes clearer that the divergence in the media positions manifests respective state narratives.

The overall meaning grounded in the data suggests that both print media have followed their agendas to address the issue of the Kartarpur project. The Indian media linking the new development with the past relations raised more concerns about peace and security and the completion of the project. On the other hand, the Pakistani print media presented the

Kartarpur project as an opportunity to break the ice for religious harmony and religious tourism. The positive development is that both the countries' print media paved the way for a new development to be expected to open further avenues for better relations in South Asia.

## Acknowledgments

The authors would like to thank all the front-line people in India and Pakistan who tried to promote religious harmony, especially regarding the Kartarpur corridor.

## Author Contributions

**Conceptualization:** Fasih Ahmed, Muhammad Mubeen.

**Data curation:** Fasih Ahmed, Muhammad Mubeen, Muhammad Nawaz.

**Formal analysis:** Fasih Ahmed, Muhammad Mubeen.

**Investigation:** Fasih Ahmed, Muhammad Mubeen, Muhammad Nawaz.

**Methodology:** Fasih Ahmed, Muhammad Mubeen, Muhammad Nawaz.

**Software:** Fasih Ahmed.

**Supervision:** Fasih Ahmed, Muhammad Nawaz.

**Visualization:** Fasih Ahmed.

**Writing – original draft:** Muhammad Mubeen.

**Writing – review & editing:** Fasih Ahmed, Muhammad Nawaz.

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
