## [Decision Letter · Decision Letter 0]

7 Sep 2021

PONE-D-21-15093Framing South Asian Politics: A Topic Modelling Approach to Indian and Pakistani English Print Media Discourses Regarding Kartarpur CorridorPLOS ONE

Dear Dr. Ahmed,

Thank you for submitting your manuscript to PLOS ONE. After careful consideration, we feel that it has merit but does not fully meet PLOS ONE’s publication criteria as it currently stands. Therefore, we invite you to submit a revised version of the manuscript that addresses the points raised during the review process.

Editor's Comments: Please be sure to address these points carefully and respond them step by step following journal's guidelinesReviewers have raised few issues in the paper and I am also with the opinion that this paper is too raw to be published and require comprehensive few rounds of revisions.English editing is required by a native english speaker (please provide with certificate of editing from a recognized agency/editing service). Write up is loaded with lot of issues right from abstract to conclusions. Please revise almost all sections of the paper for clarity, conciseness, and novelty.I am unable to find the novelty of the paper in terms of scientific research gap.A reasonable rationale is missing. Problem statement could be more convincing and obvious.Try not to use bullet points in the introduction section or anywhere until those are unavoidable.Try not to make new section/sub-section if that consists of only 1 or 2 lines.Methodology is lengthy and unorthodox. Please follow a standard version of methodology and explain study area, data collection, sampling, methods and analysis clearly one by one.Current presentation of figures and tables is too poor to grasp. Few of them are screenshots. Captions do not present complete information and formatting is awful. Adjust all of these issues. Provide sources of the figures and tables under the caption as it is done in scientific papers.Increase the canvas of analysis and provide more information in the form of tables and figures. Two tables and figures are insufficient.Third section is data analysis and 4th was supposed to be results. How can authors discuss results without presenting results? This is a blunder.How about conclusions sections? Without conclusions how one can reach to the study's findings?There are many other issues which will be highlighted in later stage of revisions as basic things needed to be inline first.Look for other issues in the paper, as there are many problems in the analysis which I did not speak of.

We look forward to receiving your revised manuscript.

Kind regards,

Ghaffar Ali, PhD

Academic Editor

PLOS ONE

Journal Requirements:

1 .Please ensure that your manuscript meets PLOS ONE's style requirements, including those for file naming. The PLOS ONE style templates can be found at

2. PLOS ONE does not copy edit accepted manuscripts (https://journals.plos.org/plosone/s/criteria-for-publication#loc-5). To that effect, please ensure that your submission is free of typos and grammatical errors.

“NO”

“NO”

5. Please amend the manuscript submission data (via Edit Submission) to include authors Muhammad Mubeen and Muhammad Nawaz.

6. Please ensure that you refer to Figure 1 & 2in your text as, if accepted, production will need this reference to link the reader to the figure.

7. We note you have included a table to which you do not refer in the text of your manuscript. Please ensure that you refer to Table 1 in your text; if accepted, production will need this reference to link the reader to the Table.

Reviewers' comments:

Reviewer's Responses to Questions

**Comments to the Author**

1. Is the manuscript technically sound, and do the data support the conclusions?

Reviewer #1: Yes

Reviewer #2: Yes

2. Has the statistical analysis been performed appropriately and rigorously? 

Reviewer #1: No

Reviewer #2: Yes

3. Have the authors made all data underlying the findings in their manuscript fully available?

Reviewer #1: Yes

Reviewer #2: Yes

4. Is the manuscript presented in an intelligible fashion and written in standard English?

Reviewer #1: Yes

Reviewer #2: No

5. Review Comments to the Author

Reviewer #1: The paper analyzes the Indian and Pakistani print media on opening of the Kartarpur corridor to facilitate the Indian Sikh community to celebrate their religious rituals. The author used LDA to find out the patterns in the print media of both countries. There are few comments and suggestions below which authors should consider to improve the article:

• The section 3 is data analysis, however, the only analysis in this section is given in figures and its description. The tables just show the keywords for search and little description. The authors should explain the figures a bit more in detail.

• Discussion section is too short. The authors concluded too soon. Based on the keywords they analyzed, they could have given more explanation followed by the current discussion.

• Overall data analysis and discussion section is short. These sections can be improved.

Reviewer #2: Title: Framing South Asian Politics: A Topic Modelling Approach to Indian and Pakistani English Print Media Discourses Regarding Kartarpur Corridor

This paper is about the divergent frames of Indian and Pakistani English print media on opening the Kartarpur corridor. The idea is novel and interesting. Structure of paper is good and comprehensive. This paper can be considered for publications in Plos One after minor revision. My suggestions for improvement are as follows;

1- The English of papers needs to be improved.

2- There are many grammatical errors in paper, for example excessive and wrong use of article “the” is present throughout text.

3- Correct use of punctuation is needed. There are many punctuation used wrongly.

4- Check formatting of tables and figures according template of journal. Table text should be aligned.

5- Use uniform formatting for all references.

6. PLOS authors have the option to publish the peer review history of their article (what does this mean?). If published, this will include your full peer review and any attached files.

Reviewer #1: No

Reviewer #2: No

---

## [Author Response · Author response to Decision Letter 0]

10 Nov 2021

Response to Reviewers’ Comments

English editing is required by a native English speaker (please provide with certificate of editing from a recognized agency/editing service). 

The article has been thoroughly proofread by the authors. In addition, it was also edited by the language expert. The editing certificate is attached as proof. 

Write up is loaded with lot of issues right from abstract to conclusions. Please revise almost all sections of the paper for clarity, conciseness, and novelty.

Thank you for highlighting this issue, all the sections of the paper have been revised and made the changes by deleting extra information and adding only relevant information concisely.

Explain clearly the novelty of the paper in terms of scientific research gap.

This paper is carried out by applying the lexical study of Natural Language Processing (NLP) through its Latent Dirichlet Allocation (LDA) tool to find out the general patterns in the print media of both countries. LDA provides several keywords and arranges them based on their weightage in the group. The word having high weightage refers to the influence of that word in the group. In addition detailed research gap has been provided (see page 4, line 3-19)

A reasonable rationale is missing. Problem statement could be more convincing and obvious.

After reviewing the previous literature, it is noted that no concrete corpus-based research has thus far been carried out using NLP techniques on the newspaper's data and media framing through the respective national print media regarding the Kartarpur corridor. Therefore, it becomes quite important to investigate how the issue of Kartarpur has been echoed on both sides of the border, which took more than 70 years to open as the media played a key role in framing the concerns and opportunities in the masses on both sides.

Try not to use bullet points in the introduction section or anywhere until those are unavoidable.

Thanks for pointing out; however, we thought, discussed and kept as it is to align with the previous discussion of the section to give clearer picture of the review

Try not to make new section/sub-section if that consists of only 1 or 2 lines.

Thanks for pointing out the issue. The subsections have been merged under one section especially in methodology section (please see page 4. Line 20-48, page 5 line 27-49)

Methodology is lengthy and unorthodox. Please follow a standard version of methodology and explain study area, data collection, sampling, methods and analysis clearly one by one.

Done accordingly by removing irrelevant and adding relevant material. 

Current presentation of figures and tables is too poor to grasp. Few of them are screenshots. Captions do not present complete information and formatting is awful. Adjust all of these issues. Provide sources of the figures and tables under the caption as it is done in scientific papers. 

Very comprehensive feedback. We have addressed all these issues accordingly. Moreover, we have conducted analysis using Python programming language. The results, produced through Python programming language have been exported in the form of images. This is the reason that all results are in the form of images. However, the quality of images has been improved.

Increase the canvas of analysis and provide more information in the form of tables and figures. Two tables and figures are insufficient. 

[Each table has been explained thoroughly. Added new figures with the explanation 

Third section is data analysis and 4th was supposed to be results. How can authors discuss results without presenting results? This is a blunder.

[Changed accordingly as Interpretation of Results with Analysis] 

How about conclusions sections? Without conclusions how one can reach to the study's findings?

 [Revised the conclusion section comprehensively]

Additional Changes

The manuscript has been formatted according to journal template guidelines.

---

## [Decision Letter · Decision Letter 1]

28 Dec 2021

PONE-D-21-15093R1Framing South Asian Politics: An Analysis of Indian and Pakistani English Print Media Discourses Regarding Kartarpur CorridorPLOS ONE

Dear Dr. Ahmed,

Thank you for submitting your manuscript to PLOS ONE. After careful consideration, we feel that it has merit but does not fully meet PLOS ONE’s publication criteria as it currently stands. Therefore, we invite you to submit a revised version of the manuscript that addresses the points raised during the review process.

We look forward to receiving your revised manuscript.

Kind regards,

Ghaffar Ali, PhD

Academic Editor

PLOS ONE

Journal Requirements:

Reviewers' comments:

Reviewer's Responses to Questions

**Comments to the Author**

1. If the authors have adequately addressed your comments raised in a previous round of review and you feel that this manuscript is now acceptable for publication, you may indicate that here to bypass the “Comments to the Author” section, enter your conflict of interest statement in the “Confidential to Editor” section, and submit your "Accept" recommendation.

Reviewer #1: (No Response)

Reviewer #2: All comments have been addressed

2. Is the manuscript technically sound, and do the data support the conclusions?

Reviewer #1: Partly

Reviewer #2: Yes

3. Has the statistical analysis been performed appropriately and rigorously? 

Reviewer #1: Yes

Reviewer #2: Yes

4. Have the authors made all data underlying the findings in their manuscript fully available?

Reviewer #1: No

Reviewer #2: No

5. Is the manuscript presented in an intelligible fashion and written in standard English?

Reviewer #1: Yes

Reviewer #2: Yes

6. Review Comments to the Author

Reviewer #1: A comprehensive interpretation of results based on figures is missing. The authors should fill this gap.

The conclusion section is too short. Although the authors described the two major concluding points in this section. However, based on the scope of this article, results and discussion still there exist room for improvement.

Reviewer #2: (No Response)

7. PLOS authors have the option to publish the peer review history of their article (what does this mean?). If published, this will include your full peer review and any attached files.

Reviewer #1: No

Reviewer #2: No

---

## [Author Response · Author response to Decision Letter 1]

27 Jan 2022

Reviewer’s Comments: A comprehensive interpretation of results based on figures is missing. The authors should fill this gap.

Authors’ Response: As per the recommendations of one the reviewers, the details about Figures has been incorporated (Please see page 9, 10, 11 and 12). Figures have been discussed in detail.

Reviewer’s Comments: The conclusion section is too short. Although the authors described the two major concluding points in this section. However, based on the scope of this article, results and discussion still there exist room for improvement. 

Authors’ Response: The conclusion section has been enlarged by adding one more paragraph based on the results and discussion. 

Additional Changes:

Some typos have been corrected throughout the manuscript. The track changes version is available as proof.

---

## [Editor Report · Decision Letter 2]

4 Feb 2022

Framing South Asian Politics: An Analysis of Indian and Pakistani English Print Media Discourses Regarding Kartarpur Corridor

PONE-D-21-15093R2

Dear Dr. Ahmed,

We’re pleased to inform you that your manuscript has been judged scientifically suitable for publication and will be formally accepted for publication once it meets all outstanding technical requirements.

Kind regards,

Ghaffar Ali, PhD

Academic Editor

PLOS ONE